# Investigation of Effect of Nozzle Numbers on Diesel Engine Performance Operated at Plateau Environment

Zhipeng Li [1,2], Qiang Zhang [1,2,*], Fujun Zhang [1], Hongbo Liang [2] and Yu Zhang [3]

1   School of Mechanical Engineering, Beijing Institute of Technology, Zhongguancun Campus,
    Beijing 100072, China; li03010301@126.com (Z.L.)
2   China North Engine Research Institute, Tianjin 300400, China
3   Department of Mechanical Engineering, Hangzhou City University, Hangzhou 310015, China
*   Correspondence: zhangqiang14020916@126.com

**Abstract:** The effect of nozzle number on the combustion and emission characteristics of diesel engines operating at high altitudes was investigated in this study. A three-dimensional computational fluid dynamics model was developed to simulate the spray spatial distribution, which is closely related to the nozzle number. The intake pressure was identified as the dominant factor under varying altitudes, while the fuel mass, injection timing and temperature were maintained constant. Altitudes of 3000 m were chosen to represent typical high-altitude conditions, and sea level cases were simulated for comparison. The results demonstrated that high-altitude operation reduced the air utility in the combustion chamber, leading to suppressed soot oxidization and worse soot emissions. Moreover, more injection nozzles will decrease the fuel injection pressure, resulting in inadequate fuel diffusion and detrimental effects on the combustion efficiency and soot control. However, too few nozzles may cause wall collisions and worsen the combustion conditions. The number of nozzles also influences the combustion, with a higher number of nozzles exacerbating poor combustion conditions. The optimal number of nozzles for the engine studied is determined to be six. Hence, determining the optimal nozzle number plays a vital role in achieving the optimal performance of highland diesel engines. This study provides valuable guidance for the development of diesel engines in high-altitude environments, where controlling the fuel consumption and soot emissions is challenging.

**Keywords:** diesel spray; combustion deterioration; numerical simulations; altitude effect; nozzles

## 1. Introduction

The diesel engine is widely used in various fields because of its high thermal efficiency and durability [1]. However, with the severe climate challenge around the world, the energy conversion efficiency and carbon dioxide emissions are receiving increasing attention. Most of these climate problems are attributed to humans and have also attracted significant attention from countries all over the world. Therefore, the world has reached a consensus that energy savings and emission reductions are vital [2]. Nowadays, more and more clean energy is gradually replacing traditional energy. The application of alternative fuels is considered an effective method [3], while improving the thermal efficiency of diesel engines under various operating conditions is also an important aspect to consider [4]. Despite the fact that diesel engines can attain an optimum efficiency of 40–50%, unfavorable conditions such as lower intake pressures and loads can result in a degradation in performance, particularly in high-altitude areas. It is noteworthy that these areas span approximately 2.5 million square kilometers globally and are home to over 6 million registered vehicles [5]. However, vehicles operating in these areas exhibit reduced efficiency and higher carbon emissions [6].

Many facets can prove that diesel engines may suffer from performance degradation, mainly including declined efficiency [7], increased fuel consumption [8] and more pollutant

emissions [9]. In addition, there are limited studies on the combustion characteristics of diesel engines at different altitudes, most of them finding that soot emissions increase as the altitude increases [10]. For example, Szedlmayer et al. [11] found that the combustion quality of diesel engines drops sharply when the operating altitude is above 3000 m, and it can be concluded that 3000 m is the threshold for soot emissions. Our previous works also proved that highland diesel engines had a decreased cylinder pressure, a lower excess air ratio, a longer ignition delay, an extended combustion duration, an increased pressure rise rate and a reduced combustion efficiency [12].

In prior research studies, Li et al. [13] have documented the occurrence of severe piston wear in diesel engines operating in high-altitude areas. Similarly, Gan et al. [14] have reported a range of engine failures in highland diesel engines, including piston overheating, cylinder bore scratches and small holes in the cylinder head. In terms of engine efficiency, Zhang et al. [15] and Benjumea et al. [16] have found that the thermal efficiency significantly decreases with increasing altitude. Furthermore, Liu et al. [17] have proposed that high-altitude conditions may lead to elevated heat losses, thereby increasing fuel consumption rates.

However, today's diesel engines can only work within a narrow altitude range, and cannot adapt to high altitude conditions [5]. Recent works have also illustrated that diesel engines at high altitudes produce more soot, which causes a significant challenge to selective catalytic reduction and might even lead to substandard emissions [18]. Furthermore, components may be affected when diesel engines work at high altitudes [19], for instance, the valve and piston may be corroded and engines may have a high emission gas temperature and turbocharger overspeed [13]. Thus, it is necessary to analyze diesel engines at high altitudes. The performance of diesel engines is mainly determined by the in-cylinder combustion situation. This combustion belongs to the diffusion combustion, and the whole process includes spray, atomization, diffusion and combustion [20]. Recent advancements in fuel injection systems [21] and combustion technologies [22] and the utilization of alternative fuels [23,24] offer promising opportunities to improve emissions and fuel economy. These innovative approaches, when combined with advanced combustion concepts, have the potential to enhance efficiency and performance. By optimizing fuel delivery, improving combustion processes and integrating cleaner alternative fuels, significant reductions in emissions and an improved fuel efficiency can be achieved [25]. Continued research and development in these areas will drive the adoption of these innovative technologies for a more sustainable future.

Spray, as the first process, plays an indispensable role in the whole process. The spray effect is closely related to the number of nozzles. When the fuel supply pressure is constant, the lower the number of nozzles, the greater the fuel injection pressure, which will strengthen the atomization diffusion effect. In addition, due to the fuel injection pressure becoming larger, the possibilities of wall collisions and the formation of fuel film on the wall are increased, which reduces the combustion efficiency greatly [26]. However, the space between the nozzles should not be too small, because the spray between them can interfere with each other [27]. From the above description, it can be seen that the number of nozzles has a complex impact on the performance of a diesel engine.

In brief, the number of nozzles plays a significant role in diesel engine performance especially at high altitudes. However, a review of previous studies on high-altitude diesel engines reveals that the majority of research has primarily investigated the impact of factors such as intake pressure and the equivalence ratio, while the influence of the number of nozzles has received limited attention. This knowledge gap hinders the optimization of diesel engines designed for high-altitude operation. Consequently, the objective of this study is to analyze the effects of varying the nozzle number on diesel engine combustion. The outcomes of this research are anticipated to provide valuable insights and guidance for the future design of high-altitude diesel engines.

The complexity of the processes within the diesel engine cylinder presents challenges for accurate analyses using traditional methods. Additionally, modifying the research

engine for experimentation can be costly and time consuming. Hence, this study utilizes three-dimensional computational fluid dynamics (CFD) simulations to provide a comprehensive understanding of in-cylinder activities, including spray, diffusion, and combustion. This advanced approach offers unparalleled insights compared to traditional methods. Moreover, the 3D CFD model dimensions are well suited to the various operating conditions required by the experiment, such as temperature, pressure, etc. Additionally, it can simulate large diesel engines as long as the computer configuration is high enough. Meanwhile, it is helpful to understand the combustion performance of diesel engines.

The following sections will first describe the diesel engine specifications. Then, the details of the numerical model and the simulated cases will then be introduced. After verifying the prediction ability of the model, this study will analyze the effect of the nozzle number on the combustion performance, and the results will provide great help for the design of high-altitude diesel engines and promote the breakthrough of clean combustion technology for highland diesel engines.

## 2. Materials and Methods

This study investigates the performance of highland diesel engines equipped with different nozzles through 3D computational fluid dynamics (CFD) simulations. Specifically, the investigated engine was a single-cylinder, four-stroke, direct injection (DI), compression ignition (CI), turbocharged and intercooled heavy-duty diesel engine, which has been modified from a commercially available turbocharged, intercooled heavy-duty diesel engine. Table 1 shows the engine specifications.

**Table 1.** Specifications of the diesel engine.

| Engine Model | |
|---|---|
| Type | 1-cylinder, 4-stroke, DI, CI |
| Bore | 150 mm |
| Stroke | 160 mm |
| Connecting rod length | 300 mm |
| Valve timing | IVO: 60 °CA BTDC |
| | IVC: 54 °CA ABDC |
| | EVO: 70 °CA BBDC |
| | EVC: 46 °CA ATDC |
| Fuel injector | Number of nozzles: 10 |
| | Nozzle hole size: 0.32 mm diameter |
| Compression ratio | 13.5 |
| Combustion chamber | Omega-bowl |
| Maximum power | 67 kW at 2200 rpm |
| Maximum torque | 320 Nm at 1500 rpm |

The numerical model for this study was established using the commercial software ANSYS/Forte. The multivariate model employed in this study has the capability to fully capture the impact of multiple variables on the combustion process within the diesel engine cylinder, making it an essential tool for analyzing diesel engine combustion. This study employs a 3D computational fluid dynamics (CFD) model to analyze the combustion characteristics of highland diesel engines with different nozzle numbers. The model focuses on the period from intake valve closing (IVC) to exhaust valve opening (EVO) to reduce computational costs. The sector grid representation effectively handles the interactions between spray holes. An adaptive mesh generation method is used to ensure simulation accuracy. n-heptane is utilized as a surrogate fuel to simulate diesel combustion due to its similar characteristics. The initial temperature of the fuel droplets is set at 400 K, and the spray development, atomization, vaporization, and mixing with air are determined by the spray model and turbulent model. At bottom dead center (BDC), the initial composition inside the cylinder consists only of air. This approach allows for the comprehensive simulation of the diesel combustion process and its interaction with the surrounding air,

providing valuable insights into the performance and emissions of diesel engines. The model accurately represents combustion behavior and emission processes by considering high- and low-temperature reaction paths. The measurements in our previous study were conducted with high accuracy and precision. The engine speed was maintained within 0.1% error using a hydraulic dynamometer. The fuel injection timing and needle valve lift were measured with Hall effect proximity sensors. The cylinder pressure, considered a dominant indicator of combustion, was tracked using a piezoelectric pressure sensor with ±0.05% accuracy. Temperature measurements were taken with K-type thermocouples, providing an error of less than 2 K. Fuel consumption and CO, NOx and UHC emissions were measured with instruments offering accuracies of ≤0.1%, 1.2 ppm, 1.5 ppm and 20 ppm, respectively, and the soot concentration was measured with an opacimeter. These accurate measurements ensure reliable data for our analysis and modeling. To minimize the variations between different cycles, pressure data were collected as the average result of 100 cycles. This approach ensures more reliable and representative measurements for our analysis. A more detailed introduction to this multivariate model can be found in reference [6]. Due to the comprehensive nature of this model, this study will only provide the most important information.

In order to accurately simulate the spray–cavity interactions and combustion process, this study employed the Renault average Navier–Stokes (RANS) method. Due to the complexity of direct numerical simulations (DNS) and large eddy simulations (LES), RANS was deemed more appropriate for this study. The renormalization (RNG) k-ε turbulence model [28] was utilized to capture the fluid motion and turbulence distribution in the Ω-shaped combustion chamber. Furthermore, a mixed Kelvin–Helmholtz/Rayleigh–Taylor (KH-RT) model [29] was employed to calculate the atomization and fragmentation behaviors of the liquid jet. The grid of the KH-RT model is primarily dependent on the relative liquid–gas velocity; thus, the unsteady gas jet model [30] was utilized to reduce the grid size dependence on the droplet–ambient gas coupling.

In order to simulate the diesel injection process, a number of models have been selected to accurately capture the relevant physical phenomena. Specifically, the injection process involves fuel catch-up and push-away, and to reduce mesh dependence during simulation, a collision mesh model [31] has been selected to simulate the collisions of liquid droplets. In addition, the study utilizes a discrete multi-component (DMC) fuel evaporation model to track the components of the actual alternative fuel during the evaporation of spray droplets. The relationship between turbulence and combustion is described using the turbulence kinetics interaction model [32], while wall film dynamics resulting from sticking, splashing or spreading impingement are simulated using O'Rourke and Amsden's wall film model [33]. To simulate the premixed combustion process and the mixing-controlled combustion process with a balance between computational cost and accuracy, this study selects the G-equation model [31]. Overall, this model plays a critical role in capturing the complex physics of diesel injection and combustion, and is chosen based on its ability to accurately represent the relevant phenomena while remaining computationally efficient. To ensure accurate measurements of soot emissions, this study utilized a two-step soot model consisting of competing formation and oxidation steps [34]. In conjunction with the basic governing equations (i.e., species conservation equation, fluid continuity equation, momentum conservation equation and energy conversation equation), this model contributed to a comprehensive representation of the three-dimensional fluid flow, the spray dynamics and the in-cylinder combustion behavior. It is essential to highlight the key assumptions adopted for the modelling process: (1) The flow motion at the intake valve closing (IVC), which marks the start of the simulation, is neglected to simplify the modeling procedure. (2) The interaction between sprays is considered insignificant and therefore disregarded in the analysis. (3) The diesel fuel is represented by n-heptane as a surrogate for modeling purposes. (4) The spray droplet diameters are assumed to follow the Rosin–Rammler distribution.

In addition to the number of nozzles, altitude plays a crucial role in this study's simulation cases. As previously stated, an altitude of 3000 m represents a threshold for combustion performance [11]. Therefore, this study aims to analyze the combustion performance of 6-, 8- and 10-nozzle diesel engines at both sea level and 3000 m altitude. The results of the analysis can provide valuable theoretical support for the future improvement of diesel engine performance.

To ensure reliability, the model was compared to experimental results [35] from a ten-jet-nozzle diesel engine. This allowed for an analysis of the model's accuracy and effectiveness in simulating the engine performance. The comparison validates the model's accuracy and confirms its capability to accurately represent the engine performance.

## 3. Results

This section will firstly investigate the macroscopic performance of diesel engines in different operating conditions through cylinder pressure, combustion efficiency and thermal efficiency. Secondly, the emission performance of a diesel engine will be analyzed through the evolution of respective in-cylinder concentrations. Finally, 3D simulation solutions will be adopted to assist revealing the specific changes in in-cylinder activities with different nozzles numbers. To validate the accuracy of the established 3D CFD model in this research, the predicted pressure trace and heat release rate (HRR) are compared with those of experiments, which are both measured at the altitude of 3000 m above sea level. As can be seen from Figure 1, the predicted pressure trace agrees well with that of the experiment, despite the slight difference in peak pressures. In addition, at the start of combustion, the combustion phases and duration of combustion reflected in the HRR curve of the CFD model are in good agreement with those of the experiments. Although the HRR traces are not completely overlapping, the established CFD model successfully captures the dominant features of the HRR trace of the high-altitude diesel engine. This indicates that the established 3D CFD model successfully simulated the in-cylinder combustion of this engine under high-altitude conditions. The experimental data presented in this study were obtained through multiple measurements, ensuring the reliability and accuracy of the results. The developed model demonstrates a good fit with the experimental data, indicating its effectiveness in capturing the underlying phenomena. Furthermore, the sub-models employed in this study have undergone extensive validation, confirming their suitability for the research objectives. These rigorous evaluations and validations strengthen the scientific integrity and robustness of the findings in this paper.

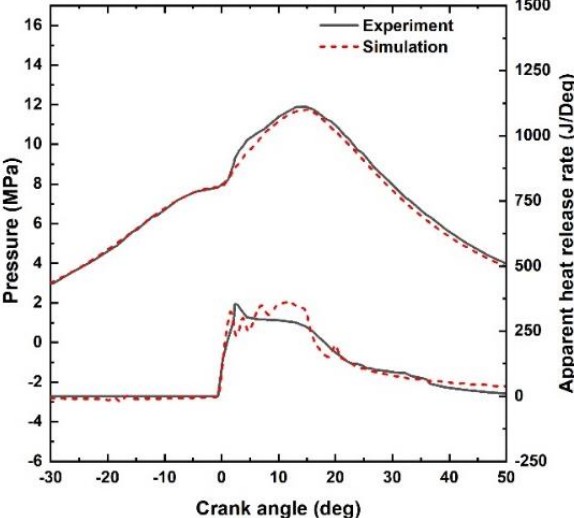

**Figure 1.** Comparison of simulated cylinder pressures and apparent heat release rates with experimental data at various altitudes (maximum torque conditions).

To investigate the impact of the number of nozzles on the overall performance of diesel engines, this study first analyzes the cylinder pressure and apparent heat release rate. As illustrated in Figure 2, it is evident that, for all three diesel engines, the cylinder pressure at sea level consistently exceeds that at an altitude of 3000 m. This can be attributed to the reduction in intake pressure in high-altitude environments, which results in a decreased air intake. Furthermore, the heat release rate curve demonstrates that higher altitudes can delay the ignition timing, i.e., the crank angle at which the heat release rate begins to increase. This leads to an increased proportion of premixed combustion and a decreased proportion of mixed controlled combustion, as well as an increase in the amount of fuel used for premixed combustion. As a result, the peak heat release rates are higher and the combustion process is rougher.

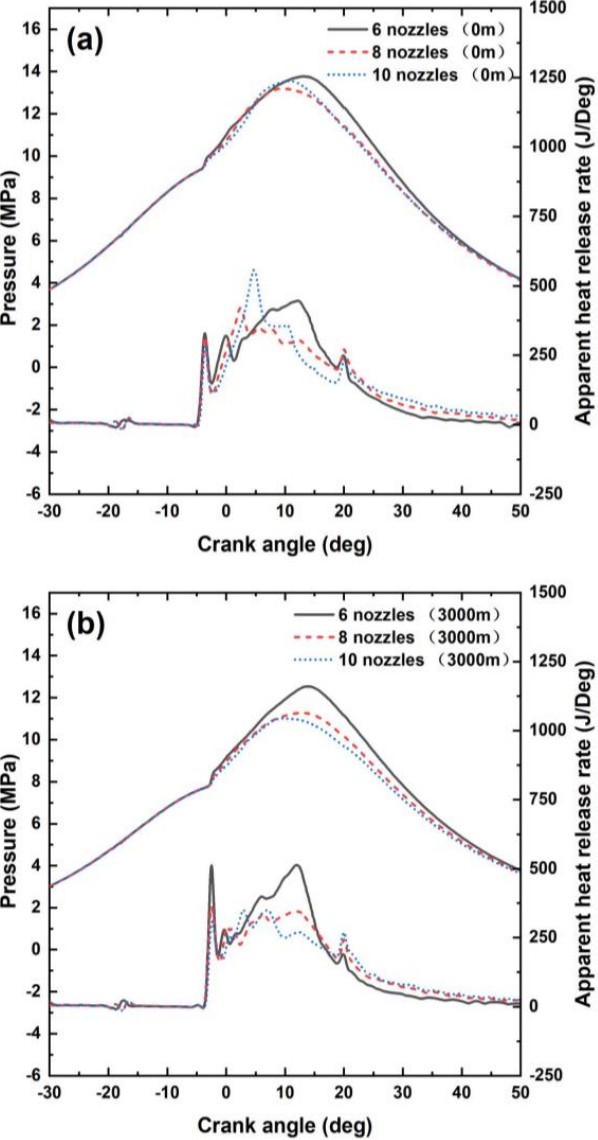

**Figure 2.** Effect of the number of nozzles on in-cylinder pressure and apparent heat release rate under maximum torque conditions at altitude of 0 m (**a**) and 3000 m (**b**).

Furthermore, the lower air pressure in the cylinder due to high-altitude conditions reduces the second peak of the heat release curve, which is unfavorable for energy conservation. With respect to the number of nozzles, the peak pressures of 6-nozzle and 10-nozzle diesel engines are similar at sea level, while the 8-nozzle diesel engines have the lowest peak pressures. At an altitude of 3000 m, the 6-nozzle diesel engine exhibits the highest

peak pressure, followed by the 8-nozzle and the 10-nozzle engines, which exhibit the lowest peak pressures. These results suggest a change in combustion phase and energy release throughout the combustion process, the causes of which require further detailed analysis. Therefore, it is necessary to further investigate the underlying reasons by a 3D simulation analysis in the following section.

Before conducting 3D simulations to investigate the impact of nozzle number on the highland diesel engine performance, it is crucial to assess diesel engine efficiency from a macroscopic perspective, which can reflect the overall engine performance. As shown in Figure 3a, the combustion efficiency of the 6-nozzle diesel engine is 98% at both sea level and an altitude of 3000 m. For the 8-nozzle and 10-nozzle diesel engines, the combustion efficiency is also 98% at low altitudes, but low combustion efficiencies occur at high altitudes.

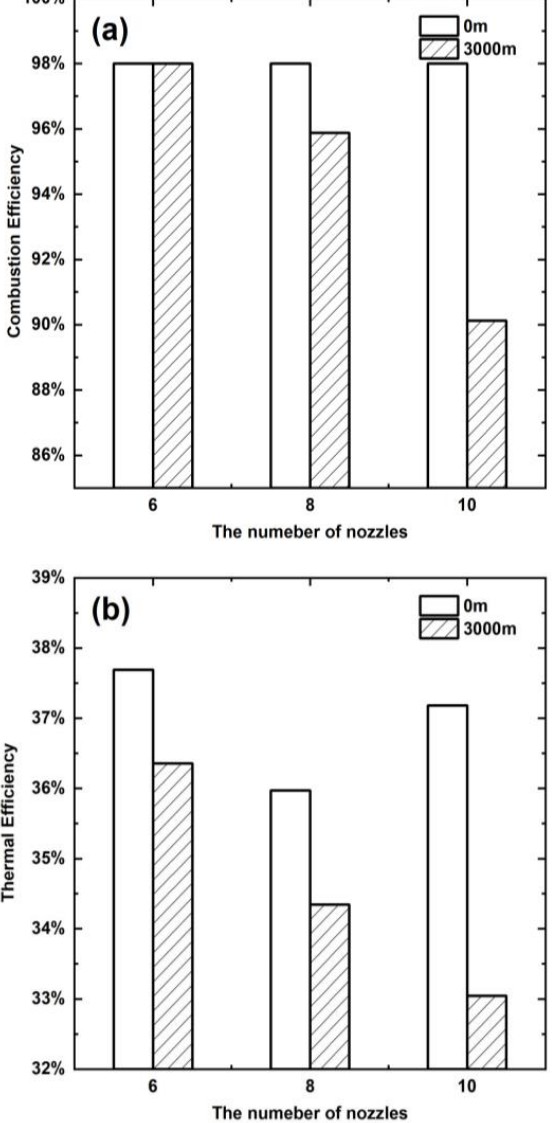

**Figure 3.** Effect of altitude and the number of nozzles on engine combustion efficiency (**a**) and thermal efficiency (**b**).

In terms of the thermal efficiency, Figure 3b shows a decrease in the thermal efficiency for all three diesel engines at high altitudes. Gan et al. [14] reached a similar conclusion, but from the perspective of an ensemble empirical mode decomposition analysis of pressure signals. The delayed ignition time inhibits the conversion of mechanical work, leading to

more negative work produced due to the increased proportion of premixed combustion. The 10-nozzle diesel engine has the lowest thermal efficiency of the three, consistent with the previous pressure profile analysis. Additionally, even though the 6-nozzle diesel engines achieve complete combustion at both sea level and 3000 m, there is a difference in their combustion phases, resulting in different thermal efficiencies at high and low altitude conditions. Specifically, high altitudes can prolong the combustion time, which is detrimental to thermal efficiency.

In addition to engine efficiency, the ability to perform external work is also a crucial indicator of engine performance. In this regard, the indicated mean effective pressure (IMEP) is commonly used to evaluate the engine's capacity to perform work, independent of engine displacement.

The ability of engine power performance is evaluated using the IMEP. As shown in Figure 4, a high altitude has a negative impact on the diesel engine's ability to perform work, with the 10-nozzle engine experiencing the most significant reduction. Furthermore, regardless of altitude, the IMEP of the 8-nozzle engine is the smallest, which is consistent with the previous thermal efficiency results and indicates the degradation of combustion.

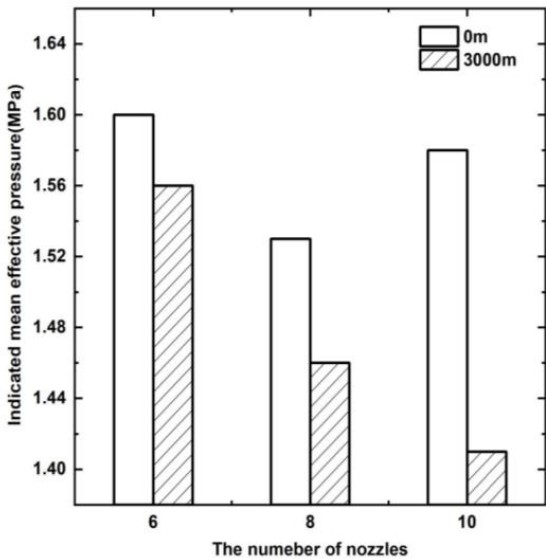

**Figure 4.** Effect of altitude and the number of nozzles on the indicated mean effective pressure.

The emissions from diesel engines are an important factor that consumers consider due to stricter emission laws. This study analyzes the mole fraction traces of soot, NOx, CO, and HC emissions in diesel engines. Figure 5a shows that the generation time of CO is nearly identical in all operating conditions, and its concentration rapidly increases to a peak. Moreover, the CO production rates are similar across different operating conditions, as indicated by the highly overlapping CO concentration peaks. However, the CO oxidation stages vary between the operating conditions. Diesel engines exhibit a lower CO consumption at high altitudes due to the reduced air content and air utilization, making CO oxidation difficult. Secondly, CO is almost completely oxidized at 0 m and 3000 m for the 6-nozzle diesel and also at 0 m for the 8-nozzle and 10-nozzle diesel, but not completely at other operating conditions. This corresponds to the combustion efficiency shown in Figure 3. The variation pattern of UHC and CO is similar, as shown in Figure 5b. The generation phase is almost the same for all operating conditions, and there are differences in the consumption phase. The main difference is the consumption rate, with diesel engines exhibiting a lower consumption rate at high altitudes. In terms of the number of nozzles, the 6-nozzle engine has the highest consumption rate, while the 10-nozzle engine has the lowest consumption rate. Overall, UHC is consumed at a faster rate than CO because the C-C bond is more easily broken.

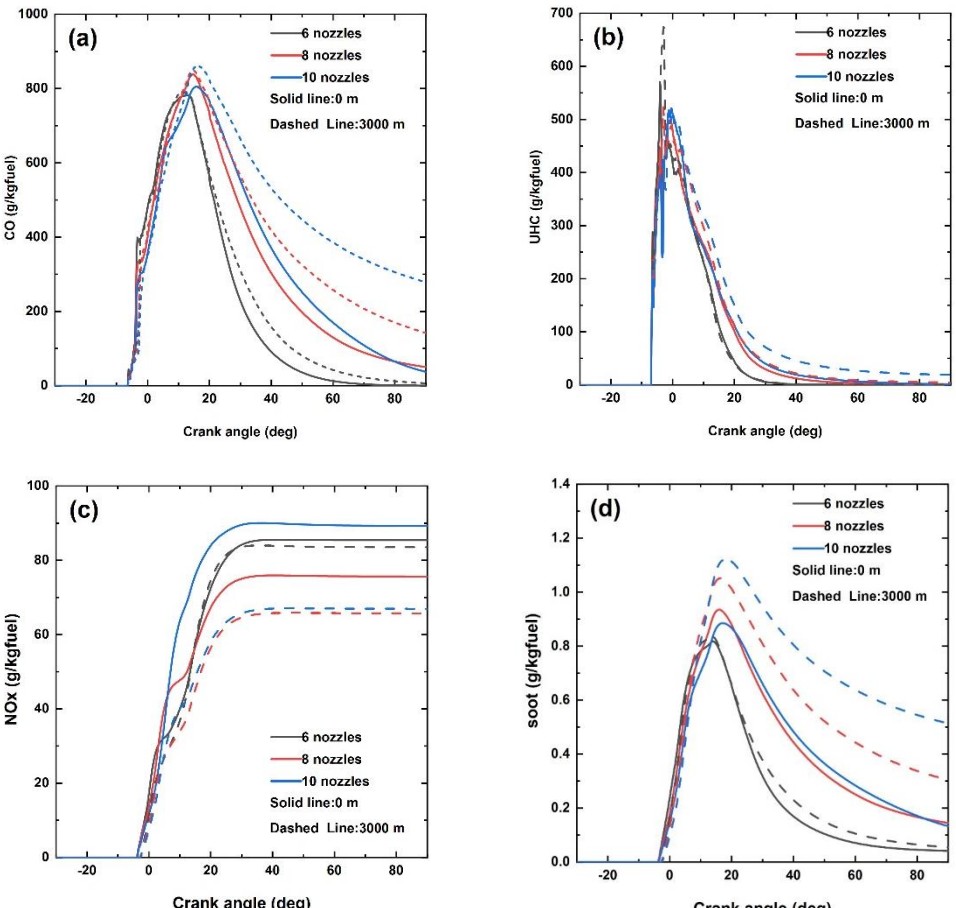

**Figure 5.** Effect of altitude and the number of nozzles on the change in average concentration of CO (**a**), UHC (**b**), NOx (**c**) and soot (**d**) in the cylinder under maximum torque conditions.

Figure 5c shows that the generation times of NOx are almost identical under each operating condition and are mainly distributed between 5 °CA BTDC and 30 °CA ATDC. These are the main exothermic processes, and are consistent with the heat release shown in Figure 2. The thermal NOx theory suggests that the formation of NOx requires a high-temperature, an oxygen-rich environment and a longer reaction time. Moreover, high altitudes can reduce the rate of NOx formation and the concentration of NOx emitted. This prolongs the burn cycle and lowers the in-cylinder temperature, which is detrimental to thermal NOx. This implies that high altitudes can have a detrimental effect on combustion since the decrease in NOx formation suggests a smaller area of thermal nitrogen monoxide (NO) production for higher altitude operations. In other words, an increase in altitude leads to a reduction in the spatial scale of the diffusion flame. In terms of the number of nozzles, at low altitude, the 10-nozzle diesel exhibited the highest NOx generation, while the 8-nozzle had the lowest. At high altitude, the 6-nozzle diesel exhibited the highest NOx generation, while the 8-nozzle had the lowest. Figure 5d illustrates the evolution of soot, which indicates that a high altitude accelerates soot formation. The number of nozzles does not significantly affect the rate of soot formation. However, in terms of peak soot, the 10-nozzle diesel produces the largest amount of soot. Regarding consumption, high altitudes can reduce the consumption of soot. Concerning the number of nozzles, the amount of soot is the least at 0 m and 3000 m for the 6-nozzle diesel engine and also at 0 m for the 8-nozzle and 10-nozzle diesel engines, which also corresponds to the combustion efficiency in Figure 2, as these four cases have high air utilization and adequate soot oxidation.

In order to analyze the mixture of diesel and air in diesel engines, this section first analyzed the distribution of the equivalence ratio in the vertical and conical sections, as shown in Figure 6.

The vertical distribution of the air/fuel ratio shown in Figure 6a indicates that an increase in the number of nozzles results in spray lag. This is because when the mass of the injected fuel is constant, a larger number of nozzles leads to a smaller average injection pressure, which reduces the fuel injection velocity. Furthermore, it is evident that the collision between the gas mixture and the piston bowl wall is more pronounced for the diesel engines with six and eight nozzles, which facilitates the subsequent mixing of fuel with more air. However, the fuel–gas mixture collision effect with the wall of the 10-nozzle diesel engine is not sufficiently apparent, causing most of the fuel to gather together without adequate diffusion and mixing with more air. Regarding height, as the height increases, the penetration length of the spray tip becomes longer and the phase advances correspondingly. As a result, the degree of spray diffusion decreases and the oxidation efficiency of the fuel ultimately deteriorates.

Regarding Figure 6b, the degree of fuel diffusion along the piston bowl wall decreases significantly with an increase in the number of nozzles, which is consistent with the results obtained in Figure 6a. This is because the fuel–gas mixture gains less velocity and therefore the flow motion is not significant enough at a lower pressure, affecting the diesel spray diffusion along the wall. Moreover, when the number of nozzles is large, the angle between two adjacent spray holes is too small, causing a corresponding interaction between sprays and forming a rich fuel mixture in the overlapping region. This exacerbates the uneven distribution of spray in high-altitude areas, thereby reducing air utilization. Concerning altitude, it can be observed that with an increase in altitude, the spray velocity advances and the shape becomes narrower. This is because in high-altitude areas, the air pressure decreases, weakening the obstruction effect on the injection and increasing the spray penetration length.

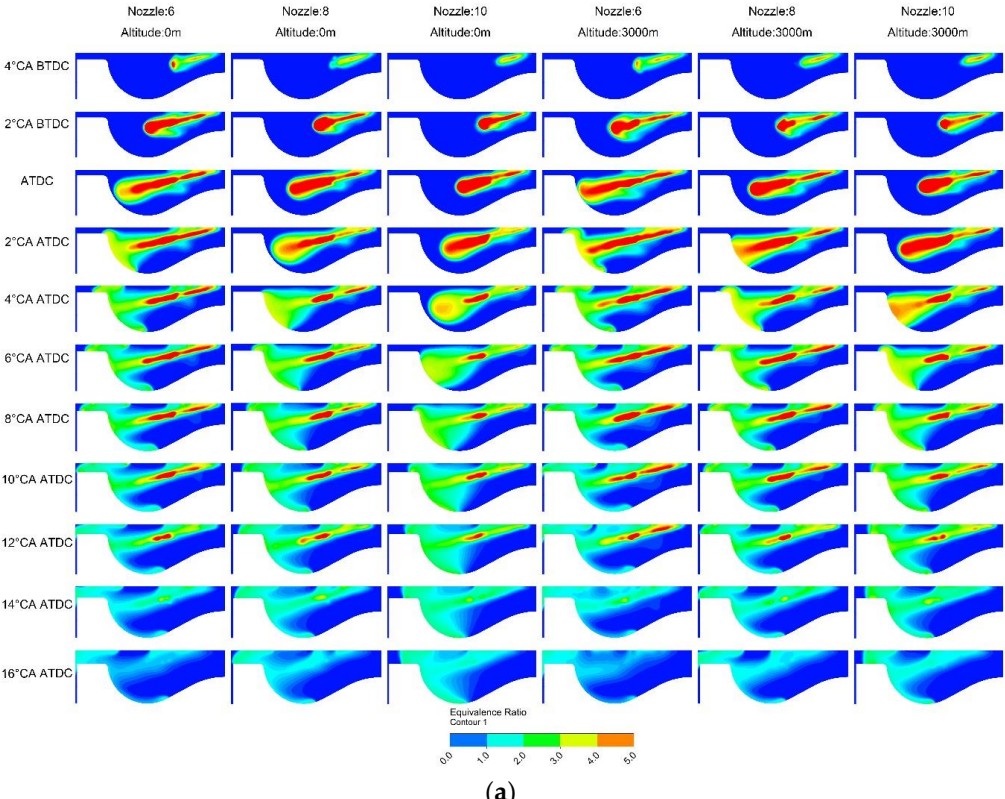

(a)

**Figure 6.** *Cont.*

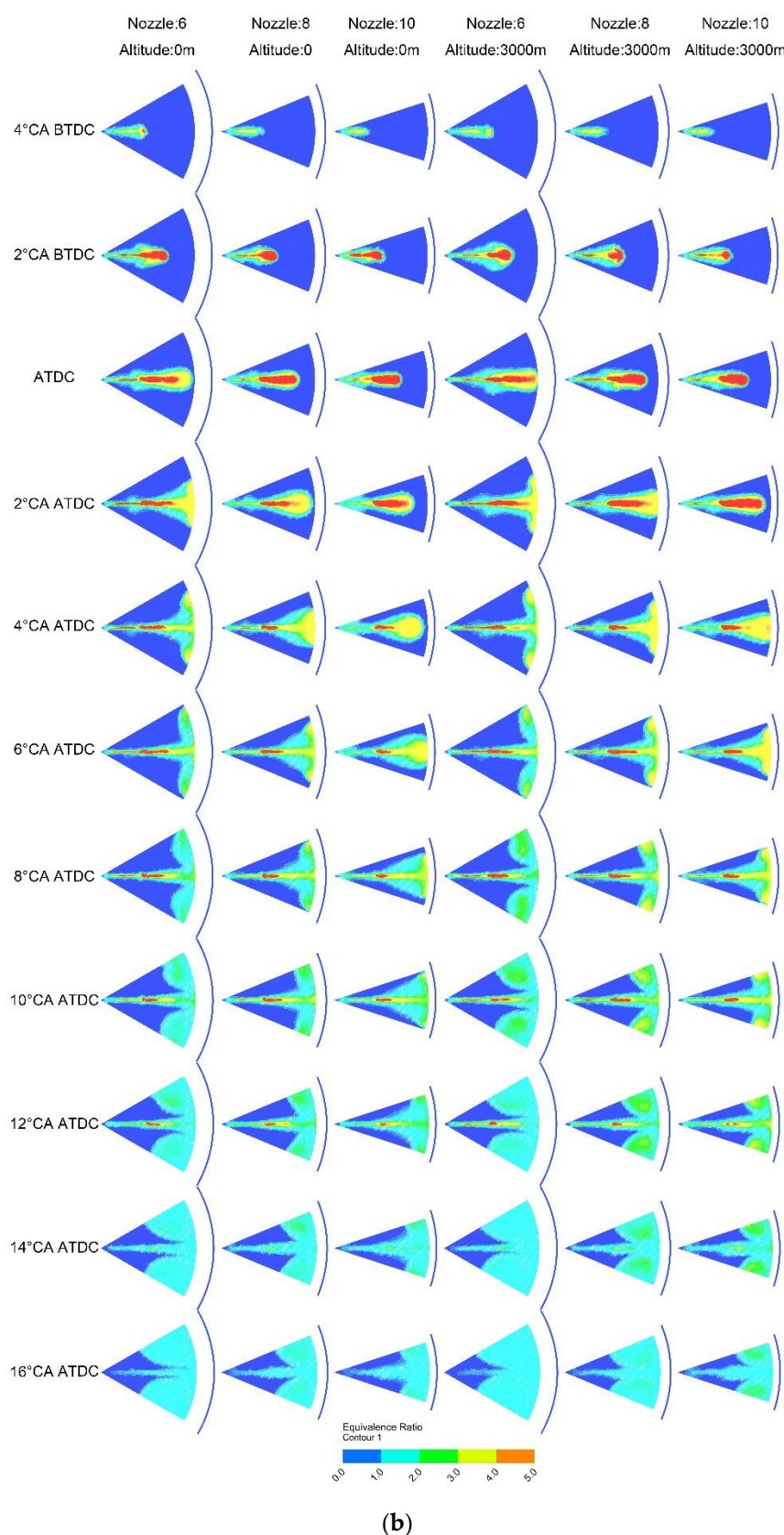

(**b**)

**Figure 6.** (**a**) Effect of altitude and the number of nozzles on the spray impinging to the combustion cavity wall on a vertical cut plane under maximum torque conditions. (**b**) Effect of altitude and the number of nozzles on the spray impinging to the combustion cavity wall on a conical cut plane under maximum torque conditions.

Regarding temperature, as shown in Figure 7a, the temperature at the front of the spray is lower than that at the rear. This is because the fuel moves from the nozzle and into the space formed by the combustion chamber and cylinder head. The front of the spray comes into contact with the air first, which results in collisions with hot air and causes the fuel to break up and evaporate. During this process, the fuel undergoes evaporation and heat absorption, leading to a lower temperature at the front end of the spray. Regarding the number of nozzles, as the number increases, a phase lag of the spray can be observed. This is due to the low injection pressure and slow injection speed, which also confirms the previously discussed air/fuel ratio distribution. In addition, with an increase in the number of nozzles, concentrated high temperature regions can be observed, which is similar to the distribution of air/fuel ratio discussed earlier. This is because the fuel diffusion effect is poor in this region, resulting in insufficient heat release reactions.

Regarding altitude, it is observed that a high altitude prolongs the ignition delay. For example, at sea level, ignition begins at −4 °CA ATDC, whereas at the corresponding altitude of 3000 m, ignition has not yet occurred. In addition, with increasing altitude, it is found that the spray shape becomes narrower. This further supports the previous discussion that the suppressive effect of high altitude on high-altitude spraying is weakened due to low pressure, resulting in a longer spray length but a narrower width for a given amount of fuel. Figure 7b also shows that with increasing altitude, the area of high-temperature regions also increases correspondingly, consistent with the previous analysis of the air/fuel ratio distribution. From Figures 6 and 7, it can be seen comprehensively that it is difficult to draw accurate conclusions from a single analysis because advancing or delaying the phase can have a significant impact on the analysis results.

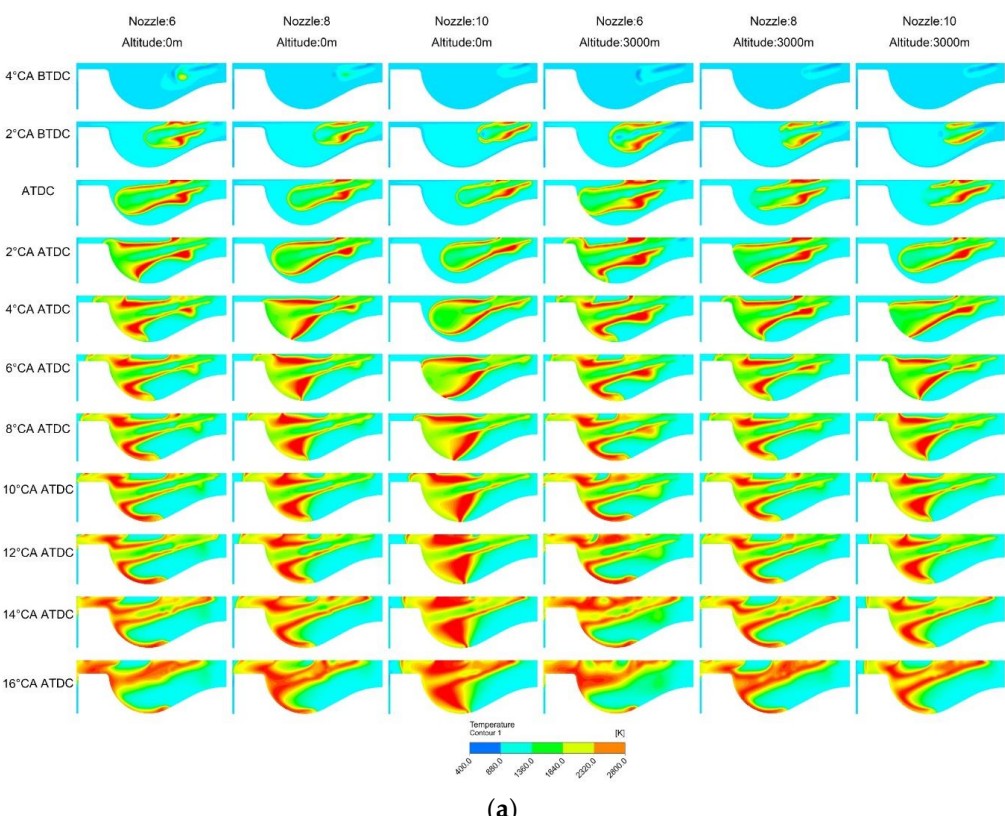

(**a**)

**Figure 7.** *Cont.*

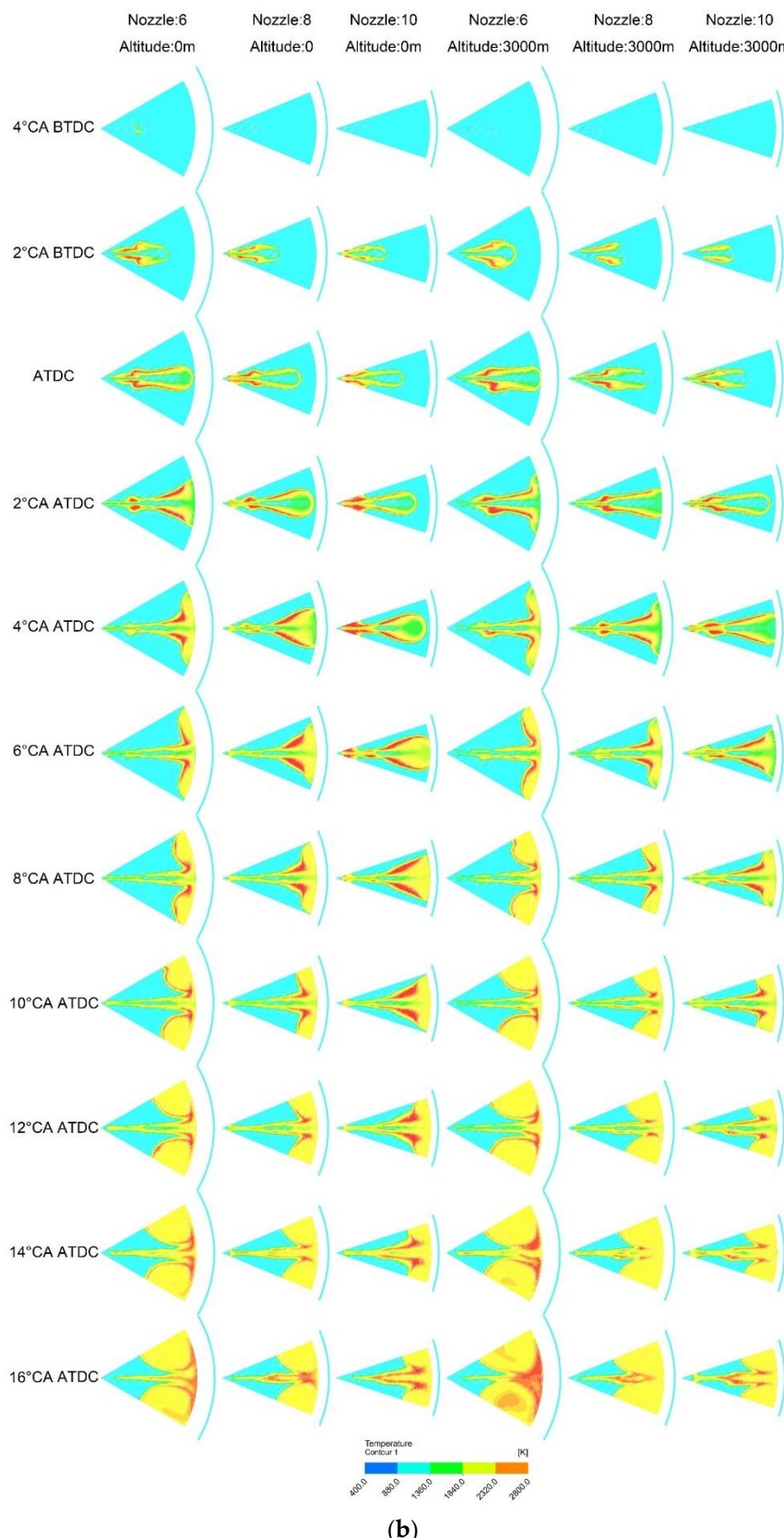

(**b**)

**Figure 7.** (**a**) Effect of altitude and the number of nozzles on the spray tip penetration on a vertical cut plane under maximum torque conditions. (**b**) Effect of altitude and the number of nozzles on the spray tip penetration on a conical cut plane under maximum torque conditions.

As we have seen from our previous studies, soot emission is an important indicator for monitoring the performance of a diesel engine. Therefore, in this study, we further analyzed the combustion characteristics of the diesel engine under different operating conditions by analyzing the spatial distribution of soot. Soot is a solid particle mainly composed of carbon, which is formed by the local high temperature, a lack of oxygen, cracking and dehydrogenation produced by combustion in the diesel engine. Here, we show the influence of the nozzle number and altitude on the spatial and temporal distribution characteristics of soot.

Figure 8a shows that the more nozzles there are, the later the generation and consumption of soot will occur, and the entire phase will be delayed. Additionally, it can be seen that in the 10-nozzle diesel engine, the amount of soot is significantly larger compared to the 6-nozzle diesel engine. In the 10-nozzle diesel engine, soot exists in the region between the combustion bowl and the near-squeeze area, whereas in the 6-nozzle diesel engine, soot exists only in the near-squeeze area. The reason for this difference is that soot formation often occurs in fuel-rich regions, which is also consistent with the results obtained from Figures 6 and 7. With regard to altitude, the spray velocity increases under high altitude conditions, which is consistent with the previous analysis because the low air pressure at high altitudes has a lower effect on the spray, causing the fuel to react prematurely with the air and form soot. This exacerbates the decrease in soot oxidation capability.

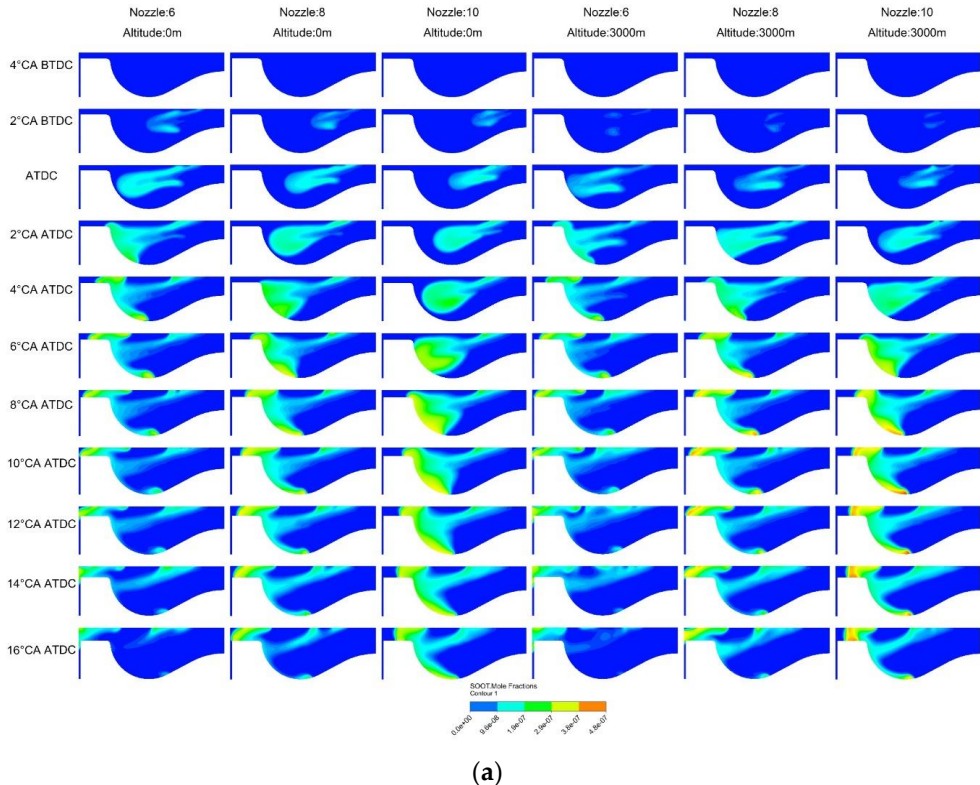

(**a**)

**Figure 8.** *Cont.*

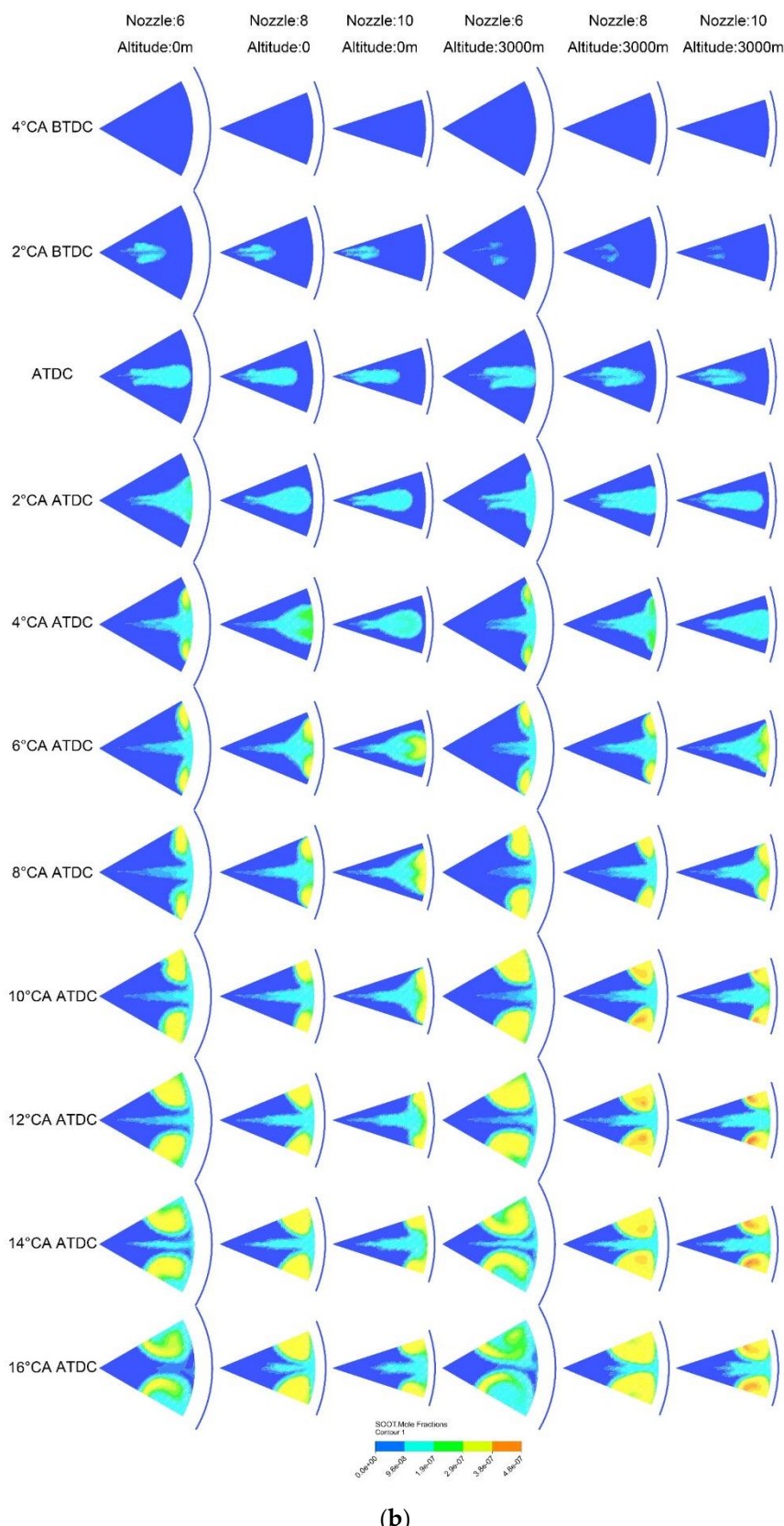

**(b)**

**Figure 8.** (**a**) Effect of altitude and the number of nozzles on the soot formation and oxidation on a vertical cut plane under maximum torque conditions. (**b**) Effect of altitude and the number of nozzles on the soot formation and oxidation on a conical cut plane under maximum torque conditions.

Moreover, it should be noted that at high altitudes, the presence of residual soot can be attributed to the insufficient oxidation caused by the limited supply of oxygen. As a consequence, residual soot primarily accumulates in the combustion zone where there is an inadequate amount of oxygen and air entrainment is restricted, resulting in an increase in the area of residual soot. Additionally, the decrease in the excess air ratio exacerbates the already compromised soot oxidation capability. This phenomenon is highlighted in Figure 8b, where the shape of soot becomes narrower with an increase in the number of nozzles due to the lower fuel pressure and diffusion. Moreover, the formation of soot is delayed owing to the small injection pressure. Notably, the higher the number of nozzles, the higher the proportion of high-concentration soot areas, which is caused by the mutual interference between nozzles. This pattern aligns with the air/fuel ratio distribution depicted in Figure 6, wherein fuel enrichment occurs in the regions of mutual interference. This state persists regardless of the altitude. Specifically, at low altitudes, the majority of generated soot is oxidized, except in the regions where spray interaction occurs, as air entrainment is prevented in these regions. However, at high altitudes, the formation of less uniform mixtures increases the formation of soot during combustion, resulting in a significant accumulation of residual soot not only in the overlapping areas but also in other regions.

In order to analyze the soot consumption in the diesel engine cylinder more accurately, this paper also presents the three-dimensional distribution of soot in the cylinder from 20 °CA ATDC to 100 °CA ATDC. From Figure 9, it can be seen that with the increase in altitude, some of the soot cannot be consumed from start to finish, and the proportion of soot increases with altitude. As for the number of nozzles, the greater the number of nozzles at high altitudes, the larger the area of soot residue. The effect of the number of nozzles on the residual soot is not very significant at low altitudes. Therefore, it can be concluded that at high altitudes, the number of nozzles should be reduced as much as possible to improve the combustion of diesel engines, but not excessively reduced, because previous studies have shown that when the number of nozzles is reduced by a certain extent, wall collision phenomena will occur, which will worsen the combustion conditions and increase the emissions of the diesel engine cylinder.

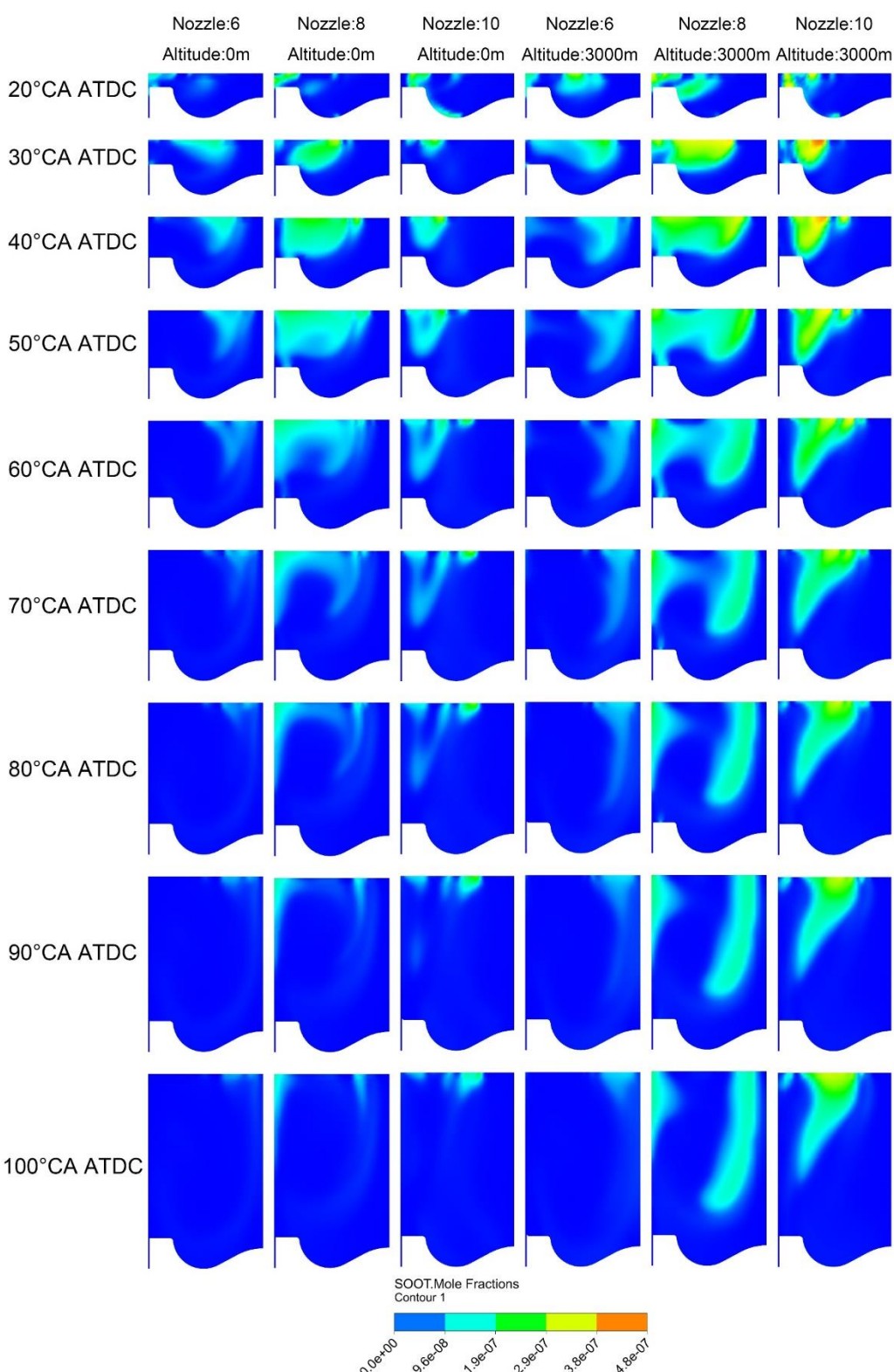

**Figure 9.** Subsequent effect of altitude and the number of nozzles on the soot formation and oxidation on a vertical cut plane under maximum torque conditions.

## 4. Discussion

From the aforementioned results, it can be inferred that high altitudes have a detrimental effect on the combustion performance of diesel engines. While the impact of nozzle numbers (6, 8 or 10) at sea level may be not significant, it becomes apparent at an altitude of

3000 m. This finding suggests that the conventional design methods employed for sea-level conditions may not be suitable for the design of diesel engines operating in high-altitude environments. Consequently, it emphasizes the importance of investigating the influence of nozzle number specifically in high-altitude conditions.

The observed phenomena can be attributed to the distinct spray distribution characteristics at different altitudes. The lower pressure prevalent at high altitudes leads to longer spray penetration, thereby magnifying the influence of the nozzle number. These findings highlight the need for a comprehensive understanding of the impact of altitude and nozzle number on the combustion performance to optimize the design and operation of diesel engines in high-altitude regions. Therefore, further research in this area is warranted to develop tailored design strategies and operational approaches that account for the unique challenges posed by high-altitude environments, ultimately leading to the improved performance and efficiency of diesel engines in such conditions. To compensate for the deteriorated spray characteristics, it is recommended to reduce the nozzle number, which would result in a higher injection velocity and improved fuel–air mixing. This, in turn, will enhance the diffusion combustion of diesel fuel and improve the thermal efficiency.

From an emissions standpoint, the nozzle number has a significant impact on CO and soot emissions. The analysis of in-cylinder combustion characteristics suggests that the decreased oxygen utilization efficiency and subsequent lower combustion efficiency are contributing factors. Additionally, the limited spray distribution at higher altitudes and increased nozzle number hinder the late oxidation process of soot, leading to a rapid increase in soot emissions. In summary, the nozzle number influences the combustion performance and emission characteristics through its impact on the spray distribution characteristics. Reducing the nozzle number can help mitigate combustion deterioration and the associated emissions. Future research efforts should focus on optimizing the nozzle design and injection parameters to achieve better spray distribution and improved combustion performance in high-altitude environments. By addressing these challenges, diesel engines can operate more efficiently and with reduced emissions in high-altitude regions.

## 5. Conclusions

This study established a 3D CFD model to investigate the effects of altitude and the number of nozzles on the combustion degradation in diesel engines under high-altitude operating conditions. The main findings were as follows.

The study found that the altitude significantly influenced the combustion of diesel engines. At altitudes above 3000 m, the engine performance deteriorated due to a reduce in-cylinder pressure, leading to less effective fuel diffusion, limited spray distribution, increased soot formation and decreased soot oxidation caused by restricted air entrainment. Specifically, with a nozzle number of 10, the thermal efficiency decreased by 12.6% as the altitude increased from sea level to 3000 m. The IMEP experienced a reduction of 10.7%, and the combustion efficiency decreased by 8.7%.

Furthermore, the number of nozzles had a notable impact on diesel combustion, particularly in high-altitude conditions. Increasing the number of nozzles resulted in degraded combustion due to reduced fuel injection capacity, decreased diffusion and increased overlap and interference between sprays. This led to a fuel-rich mixture in the overlapping area, further exacerbating poor combustion conditions. At an altitude of 3000 m, the combustion efficiency, IMEP and thermal efficiency of the diesel engine were found to decrease by 8.1%, 8.9% and 8.3%, respectively, as the number of nozzles increased from 6 to 10. These results indicate that a lower nozzle number contributes to an improved combustion performance and overall engine performance in the investigated scenario.

Overall, both the altitude and the number of nozzles are important factors influencing the combustion conditions of diesel engines. Based on the findings of this research, it was determined that the optimal number of nozzles for the engine under investigation is six. The outcomes of this research are expected to bridge the existing research gap regarding the influence of nozzle number on high-altitude diesel engines. Moreover, the findings will

offer valuable insights and guidance for the future design and optimization of such engines in high-altitude conditions.

Based on the findings of this study, it is possible to utilize analytical techniques such as machine learning to optimize the number and distribution of nozzles in diesel engine combustion chambers of different shapes. By employing these advanced analytical methods, it will become feasible to enhance the performance and efficiency of diesel engines. This approach holds great potential for future research and development in the field of combustion chamber design. However, it is important to consider challenges such as collecting and utilizing large datasets, ensuring the generalization performance of the models and addressing the computational requirements for implementing these advanced analytical techniques.

**Author Contributions:** Z.L., conceptualization, methodology, simulation and writing—draft preparation; Q.Z., validation, methodology and simulation; F.Z., validation, simulation and supervision; H.L., analysis and simulation; Y.Z., analysis and supervision. All authors have read and agreed to the published version of the manuscript.

**Funding:** This research received no external funding.

**Data Availability Statement:** The data presented in this study are available on request from the corresponding author.

**Conflicts of Interest:** The authors declare no conflict of interest.

## Abbreviations

| | |
|---|---|
| 3D | Three-dimensional |
| ATDC | After Top Dead Center |
| BTDC | Before Top Dead Center |
| CA | Crank Angle |
| CFD | Computation Fluid Dynamics |
| EVO | Exhaust Valve Opening |
| IVC | Intake Valve Closing |
| $NO_X$ | Nitrogen Oxides |
| UHC | Unburnt Hydrocarbon |

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
