# Peer review of "Investigation of Effect of Nozzle Numbers on Diesel Engine Performance Operated at Plateau Environment"

_sustainability, doi:10.3390/su15118561_

Round 1
Reviewer 1 Report
The study presented in the manuscript is related to a 3D CFD model analysis to investigate the effect of altitude and the number of nozzles on combustion degradation in diesel engines under high-altitude operating conditions. The topic is relevant and fully corresponds to the subject matter of the Sustainability journal.
The material of the article is presented in a logical sequence; in general, adhering to the recommendations of the journal, the research results are highlighted clearly and reasonably. At the same time, some separate structural additions and a number of clarifications of technological aspects of the article material should be made:
- The abstract section is too long and it should be revised reporting only the main key messages avoiding repetition or some information that can be found easily in the test methodology section. It should be about 150-250 words with concise text in a single paragraph. Answer the questions: What problem did you study and why is it important? What methods did you use? What were your main results? And what conclusions can you draw from your results?
- Most of the ideas written were already described in many literatures. The Authors tried to compile it but lack of the enhancement of the interrelation analysis between the references. It is advised that the authors give a deeper analysis of how these ideas become more applicative strategies so that they can contribute to the next step of implementation.
- Novelty: The authors partly stated what motivated the idea portrayed in this study, what then does this study offer beyond the recent advances made on the combined approaches? I would advise the authors to carefully carry out a close comparison as well in the discussion section to enumerate the advantages this study offers over other related works.
- Introduction doesn't provide a great overview of the topic. It misses the aim of the study or what is going to be done in this study. Please state clearly the aims and what this study does. However, highlight that innovative technologies in the field of fuel injection systems, combustion and combined with alternative fuels are able to improve emissions and fuel economy (SAE 2019-24-0116, SAE 2018-01-1697) in combination with advanced combustion concept and alternative fuels (10.1080/15567036.2022.2124326; 10.1007/978-981-16-8751-8_3; 10.1299/jmsesdm.2017.9.C308). These technologies/fuels could be adopted to improve efficiency and performance.
- More in-depth analysis of the author's contribution to this paper in the introduction section. I would like to see more discussion of the literature so that I can clearly identify the article relates to competing ideas.
- The results of the study are not to a small extent dependent on the accuracy class of the measuring equipment used. Please complete the methodology section with appropriate information.
- How many repeat experiments were performed at each point? Comment on repeatability.
- The language of the manuscript is fair; I would advise consulting a language editor to further polish the language of the manuscript. There are several grammatical mistakes. Please work closely with a native English speaker to refine the language of this paper.
- Challenges and future directions to improve and implement these fuel technologies with big data analytics should be discussed.
- In my opinion, there are several up-to-date approaches to the idea. Authors should look at these approaches, compare the results and prove their idea. This is a major concern.
- The conclusions don't tie to the discussion well and should be reconsidered. There needs to be a clearer discussion of the points in the body, or the conclusions should be adjusted to better match the existing discussion
- How did the authors establish/estimate the uncertainties in the measurements documented in this research?
- Author credit role: Revise authors' roles: what is the meaning of experimental preparation? Please be more explicit here as the author's roles are not clearly specified.
- Please remove this sentence: This section is not mandatory but can be added to the manuscript if the discussion is 401 unusually long or complex.
- Conclusions section: The conclusions section is too short, please consider emphasizing this section, and clarify your message this could give an impact on the paper's archival.
- Many of the paragraphs are long. It would aid the reader to have them broken into more digestible chunks.
- One very crucial thing the authors left out is that they did not discuss well the methodology and results improving scientific soundness. Without presenting a detailed discussion, the authors leave one with doubt as to whether the study was carried out.
Reviewer 2 Report
The research presents an interesting approach in accordance with the journal's scope. It is overall well structured and organized. There are only some small aspects to improve:
abstract: line 13 - I would prefer to use diesel or fuel injection rather than oil injection.
1-Introduction: it would be interesting to present some similar works that deal with internal combustion engine simulations and highlight the merits and differentiations of this research compared with that ones.
2- materials and methods: line 107 - Please discuss if the fact of having the representation of a partial analysis (60, 45, and 36 degrees) can not hide the possible interaction of the jets originating from adjacent nozzles.
4- Summary: The first paragraph can be erased. Future work can be better described and the overall merits f the work can be better presented, mainly focusing on the differentiation with similar works.
Reviewer 3 Report
The article presents an investigation of effect of nozzle numbers on diesel engine in plateau environment. The article is well written and well structured. The study was carried out on the basis of mathematical modeling. The presentation of the material is logical. The quality of the drawings is high. There are several comments on the article.
Questions and comments on the article:
1. I believe that the "Notation and Abbreviations" section of the article is necessary for the convenience of readers.
2. It is necessary to indicate the boundaries for applying the results of the study (engine size, engine type, engine assignment, operating modes, etc.).
3. Please indicate the limits of application of the mathematical model (limiting dimensions, temperatures, pressures, working environment, and so on).
4. Please indicate the amount of improvement in the accuracy of the proposed model compared to the data of other authors.
5. It is necessary to emphasize the scientific contribution of the mathematical model to the improvement of engineering methods for calculating the nozzle and improving its design.
6. Please specify the properties of the working medium in the study in more detail.
I believe that these comments should be corrected before the article is published in the Sustainability.
Reviewer 4 Report
The work entitled "Investigation of effect of nozzle numbers on diesel engine in 1 plateau environment" covers the issue of the impact of the place of use of vehicles and devices on the processes taking place in a compression ignition engine and the emissions it generates. The work contains a well-described experiment, however, it requires the following remarks to be taken into account:
1. The title of the thesis clearly indicates the influence of the number of nozzles on the engine operation. However, there is no reference to this in the abstract. Similarly, there are no clear conclusions from the content of the work in the context of the title of the work.
2. Abstract too general and not specific enough. It should describe in more detail the most important assumptions of the work. It should contain information about the height of use of the engines, or a specific indication, as a conclusion, what number of injector nozzles is optimal.
3. In addition, the abstract indicated that the work is a kind of instruction, but it is not clear from the summary of the work - see point 12.
4. The work contains many stylistic errors that need to be reformatted, e.g. line 13, 36, 56.
5. Please divide the Introduction into subject areas separated by paragraphs. There is no clearly defined purpose of the work. Lines 81-86 - this is a summary of the work, from which there is no clear purpose. I suggest transforming this paragraph with separating the purpose of the work.
6. Line 98 - not every reader has access to ref. 20 and carefully analyze the adopted model. The description of the model should be extended.
7. Please explain why 6, 8 and 10 nozzles have been adopted.
8. Please explain how the submitted work differs from the ref. 2.
9. Please explain all abbreviations and symbols used.
10. What does "maximum torque condition" mean? Have checks been undertaken in what real conditions the engines are operated at high altitudes?
11. Discussions with the works of other authors are missing in the final part.
12. Summary section - please change this section to Conclusions and extract detailed conclusions resulting from the work. An indication of what number of nozzles would be appropriate should be given.
13. Links are unformatted uniformly and contain many omissions. Please check. Otherwise items 9 and 13 are identical.
Reviewer 5 Report
Dear Authors,
The article is interesting and presents an interesting approach to the subject, but it is hardly readable due to the lack of transparency.
In my opinion, the description of the methodology should be improved because, in its current form, the chapter "Materials and Methods" is not very clear. Assumptions adopted for modelling should be presented in points.
In addition, in my opinion, the chapter "Results" contains both a description of the method, e.g. the text from lines 160-165, and a discussion of the results, e.g. line 215 - "Gan et al. [31] reached a similar conclusion,…" The "Results" chapter should be reconstructed to present the result of the obtained simulation along with a short description. The description of the method should be moved to the "Materials and Methods" chapter.
The article also lacks a "Discussion" chapter, while some parts of the "Results" chapter can be treated as a discussion. Therefore, in my opinion, the article should be restructured to include separate chapters: Materials and Methods, Results, Discussion and Conclusions.
Round 2
Reviewer 1 Report
The authors improved the manuscript and the manuscript can be considered for publication.
Reviewer 3 Report
All comments have been corrected. The article may be accepted for publication.
Reviewer 5 Report
The authors improved the manuscript and the manuscript can be considered for publication.